# Is the Parwarish parenting intervention feasible and relevant for young people and parents in diverse settings in India? A mixed methods process evaluation

Kaaren Mathias [ID] ,[1,2] Prabhudutt Nayak,[3] Pratibha Singh,[2] Pooja Pillai,[4] Isabel Goicolea[1]

¹Global Health and Epidemiology, Umea University, Umea, Sweden
²Community Health and Development, Emmanuel Hospital Association, New Delhi, Delhi, India
³Chhatarpur Christian Hospital, Community Health and Development Programme, Emmanuel Hospital Association, Chhatarpur, Madhya Pradesh, India
⁴Burans, Herbertpur Christian Hospital, Emmanuel Hospital Association, Dehradun, Uttarakhand, India

**Correspondence to**
Dr Kaaren Mathias;
kaaren.mathias@umu.se

## ABSTRACT

**Objective** To assess the feasibility, acceptability and relevance of the Parwarish, a positive parenting intervention (adapted from PLH-Teens) in three diverse settings in India.

**Design** This mixed methods study used the Medical Research Council framework for process evaluations of complex interventions.

**Setting** This study was set in disadvantaged communities in urban Agra, rural Uttar Pradesh and tribal Jharkhand in India.

**Participants** Data were collected from 86 facilitators, implementers, parents and teens who participated in the Parwarish intervention among 239 families.

**Intervention** Couples from target communities facilitated groups of parents and teens over the 14-module structured, interactive Parwarish intervention which focused on building communication, reducing harsh parenting and building family budgeting skills.

**Outcome measures** We assessed relevance, acceptability and feasibility of the intervention using mixed methods. Qualitative data collected included semistructured interviews and focus group discussions with implementers, facilitators, parents and young people who were transcribed, translated and thematically analysed to develop themes inherent in the data. Quantitative data which assessed attendance, fidelity to the intervention and facilitator training and coaching were analysed descriptively.

**Results** Findings were grouped under the three domains of facilitation, community engagement and programme support with the following seven themes: (1) community-based facilitators increased contextual validity of the intervention; (2) gender relations were not only influenced by Parwarish implementation but were also influenced and transformed by Parwarish; (3) facilitator responsiveness to group concerns increased participation; (4) participation gathered momentum; (5) Parwarish's strong core and porous periphery allowed adaptations to local contexts; (6) technology that included Skype and WhatsApp enhanced implementation and (7) critical reflection with community trained coaches strengthened facilitation quality and programme fidelity.

**Conclusion** This study found Parwarish engaging, feasible and acceptable in three diverse, low-income communities, although constrained by patriarchal gender relations. It

## Strengths and limitations of this study

► This study is set in three diverse settings providing greater generalisability.
► Research conducted by programme implementers allows close-up understanding of the context.
► There is a risk of social desirability bias in responses from participants due to research being conducted by some programme implementers.
► The 'real-world' study design increases generalisability of the findings.
► The 'real-world' study design meant the intervention was adapted to different contexts, reducing fidelity to the programme.

paves the way for larger-scale implementation in other South Asian settings.

## INTRODUCTION

Young people in India experience high rates of physical, sexual and emotional abuse as well as neglect[1 2] with a recent study estimating that 70% of adolescent girls in Delhi have experienced violence at some time in their lives.[3] India has more adolescents than any other country and the millions affected by violence risk current and life-long problems with physical and mental health, well-being, education and employment.[1 4 5] India ranked 140th globally in the latest Global Gender Gap Report[6] and gender inequality is a key driver of gender-based and family violence in India.[1]

Positive parenting interventions can potentially reduce violence in two key ways: first by promoting parenting strategies which reduce physically or emotionally violent actions, and second by reducing violent or antisocial behaviour among children.[7] Parenting in India has not been studied widely and a large systematic review examining interventions to reduce harsh parenting or promote positive

parenting in low-income and middle-income countries could not identify any studies located in India.[7] Studies examining parenting of adolescents in middle-class urban settings identify parenting is often authoritarian, extends until early adulthood, with a focus on educational attainment, and typically includes a high level of monitoring of the social behaviour of their teenagers (particularly young women).[8–10] However, we could find no interventional studies promoting positive parenting among disadvantaged adolescents in India.

There is growing evidence that interventions focusing on positive parenting can reduce abuse of young people and children.[11–13] Further, the most effective programmes are participatory, address underlying risk factors such as gender relations and support reduced non-violent behaviours, greater communication and shared decision-making among family members. Yet globally, 10% or fewer of adolescents can access violence prevention programmes.[4 14]

Parenting for Lifelong Health is a positive parenting group intervention developed in South Africa for parents and adolescents that builds on social learning and parent management training principles seeking reduce all types of violence in families.[15] Other publications detail PLH-Teens' development, theory and effectiveness.[12–16] PLH-Teens was adapted for the Indian setting by the Emmanuel Hospital Association (EHA) (EHA is a non-profit organisation working across north and North-East India with charitable hospitals and community health and development programmes) working in consultation with a group of mothers of adolescents (experts by experience) from disadvantaged communities in Dehradun, Uttarakhand. Together they reviewed both the superficial structures (language, role-plays) as well as cultural content and named the adapted intervention Parwarish (meaning 'nurturing' in Hindi). This paper reports a process evaluation of Parwarish in three diverse communities in central and north India.

Process evaluations are valuable to assess the fidelity and quality of implementation, to clarify causal mechanisms and identify contextual factors associated with variation in outcomes and to build interventions that are more generalisable and ultimately scalable.[17–19] Examining feasibility assesses whether an intervention is relevant and acceptable and how it can be adapted contextually.[20] There is very limited research documenting implementation of family violence prevention programmes, particularly in community and low-income and middle-income settings.[4 5 7 21] Many describe the urgent need for studies that examine context and implementation processes in order to better develop interventions that reduce violence in communities, particularly in low-income and middle-income settings.[4 5 7 11 19 21]

This study aimed to examine whether the Parwarish parenting intervention was feasible, relevant and acceptable for parents and adolescents and community facilitators living in three diverse and disadvantaged communities in central and northern India, in order to establish the utility of this intervention for scaled implementation across other South Asian settings.

## METHODS
### Setting of the intervention
This study examined Parwarish implementation in Agra, a large city in Uttar Pradesh, in Robertsganj, a small town in the rural Sonbhadra district, Uttar Pradesh and among indigenous people in Khunti, Jharkhand, a rural and remote setting. The context of the intervention is detailed further in the results section.

### Intervention
#### Intervention content
Parwarish seeks to reduce harsh parenting and violence within families through new attitudes and skill building between parents and adolescents. Module content includes spending quality time together, communicating positively, managing strong emotions such as anger and finding safe support when needed as well as problem-solving, conflict resolution and finance skills, including how to manage a household budget and uses role-plays, activity-based learning and home rehearsal and activities building on social learning.[15 16 22 23] Further detail of the 14 modules of the Parwarish curriculum is provided in online supplemental table 1.

### Implementation
Parwarish was implemented by the EHA Community Health and Development Programme teams based in Agra, Robertsganj (UP) and Khunti (Jharkhand). The budget for both implementation and evaluation was limited and therefore we conducted the programme using in-kind support by EHA's existing resources and team members. A coach was appointed and trained for each location and took responsibility for recruiting facilitators as well as training and coaching facilitators. Separately, a project officer at each location was responsible for research components of the project and supported baseline and endline data collection as well as monitoring and evaluation of Parwarish sessions with other EHA community coordinators in the team.

Implementation of Parwarish groups was conducted by pairs of community facilitators with the following criteria for facilitation selection:

▸ Parents of adolescents who were resident in the target community.
▸ Represent an equal mix of genders willing to work as a pair in facilitation (over half of facilitators worked as a married couple).
▸ Trusted and accepted as a leader by the community.
▸ Effective communicators.
▸ Had at least passed class 10th and were fluent in the local dialect or language.

### Parwarish training and coaching
Trainers from PLH-Teens South Africa facilitated a 10-day course for Parwarish facilitators and a 3-day training for

coaches (total n=25 participants). The facilitators then led the 14 Parwarish modules with groups of parents and teens with meetings of 1.5–2 hours and encouraged participants to complete the weekly activity to try at home, for example, family eats dinner together. Home visits with those who missed a session review the topic for the week. In each site, a local 'coach' supported facilitators with weekly meetings reflecting on their facilitation, while a fortnightly coach-the-coaches meeting was led online with someone from PLH-Teens South Africa.

### Recruitment of target communities and participants

EHA Community Health and Development team members with support from community volunteers recruited families in communities where they had been running Community Health and Development Programmes in the previous 5 or more years, by informing them about Parwarish and inviting participation. They held small community meetings informing about the proposed programme and invited any families who met the inclusion criteria and were keen to participate to give their names and details to a team member. Inclusion criteria were:

▸ At least one adolescent (aged 12–18 years) in the household.
▸ Permanent resident in the target community.
▸ At least one parent/carer and one adolescent available for weekly sessions.

A total of 239 of 245 families invited agreed to join the programme (97.6%). Based on team capacity, there were 129 families recruited from Agra, 60 families from Robertsganj and 50 families from Khunti.

### Data collection

In-depth interviews and focus group discussions (FGDs) were collected face to face in the community by KM, PS and PN in Hindi with 86 purposively sampled facilitators, parents and young people and continued until data saturation was reached. They lasted between 30 and 70 min and were recorded, translated and transcribed into English. Interview guides were developed by authors and piloted in the community. They asked about implementation processes, acceptability and feasibility as well as FGDs with parents, adolescents and facilitators. Additional field notes and free text were recorded in coach and facilitator's documents. Further data were collected during meetings with coaches which were conducted online, in the form of written notes taken during meetings by a researcher not involved in meeting facilitation or participation (KM). Online supplemental table 2 provides an overview of the qualitative data collection.

### MEASURES OF FIDELITY

Coaches also filled a paper register at each site on the following measures:

▸ Facilitation and fidelity: 14 of the 14 sessions self-reported by facilitators using a five-point checklist where a score of 1 indicated poor quality and a score of 5 indicated high quality. The following components of the intervention were assessed with this scale: physical exercise, emotional check in, main teaching topic, role-play and completion of home activity.
▸ Facilitation and fidelity: 4 of the 14 sessions for each Parwarish group were observed and measured by the coach, EHA project manager or a researcher who had all been trained to fill using the same criteria. Fidelity assessed the five aspects above and two additional aspects, namely facilitator engagement and completion of registers.
▸ Attendance of facilitators and coaches at trainings and coaching: attendance register.
▸ Attendance of parents and adolescents: attendance register.
▸ Completion of assigned home activities: register.
▸ Home visits completed by facilitators: register.

### Conceptual framework and analysis

This mixed methods study was analysed with a merging of two evaluation frameworks, namely process and feasibility evaluation, as we sought to understand how the intervention was delivered and also whether it was acceptable. We followed the Medical Research Council (MRC) framework for process evaluation of complex interventions examining how the intervention was delivered and the emerging adaptations of the intervention related to context and actors.[17] We also assessed feasibility of facets of implementation that include practical, pragmatic and implementation processes used to deliver Parwarish programme to participants, adaptations to the Parwarish intervention required to increase engagement, understanding and relevance to diverse communities. Additionally, we examined acceptability and demand for Parwarish by asking participants whether they were comfortable with the group implementation processes and programme content.[20]

Qualitative interviews and written documents were read and reread and analysed using thematic analysis developed by Braun and Clarke.[24] Four transcripts were inductively coded by PN and KM and these codes were reviewed to develop a coding framework, paying attention to implementation processes, feasibility and barriers and enablers of the intervention. Remaining transcripts were then coded with this framework by KM and PN, who collaboratively developed categories and then themes, looking for patterns inherent in the data. Given the geographically dispersed field sites data were not member checked with parents and children in communities. Quantitative data were analysed descriptively and tabulated.

### Patient and public involvement statement

Parents were involved in this study in three key ways. First, parents were the impetus for this Parwarish intervention, having requested EHA team members in Dehradun, Agra and Khunti to support them to build their skills in raising young people. Second, parents were part of

the process of adaptation of the intervention. A group of parents who resided in informal urban communities in Dehradun were consulted as 'experts by experience' in reviewing and suggesting contextually relevant adaptations of Parwarish. Third, parents from intervention target communities were identified and trained as facilitators of this intervention and thus were also involved in recruiting participants and collecting data linked to implementation required by facilitators such as the highlights and lowlights register. Parents in communities will also be involved in sharing findings at dissemination meetings planned after COVID-19 restrictions are lifted later in 2022 using community meetings and reviewing a plain-language Hindi summary of study findings.

### Stance and ethics

This study was conducted by participant researchers who had all lived in India for at least 24 years and spoke fluent Hindi except IG. KM, PN, PS and PP were employed by the implementing organisation, EHA. KM (PhD) facilitated adaptation of PLH-Teens to Parwarish and led meetings with experts by experience. PS and PN (Masters in Public Health and Masters in Social Work) supported Parwarish implementation at a high level. PP (M.MedSci) and IG (PhD) are women who work in India and Sweden, respectively.

### Findings

Findings are presented under the broad headings of Context and Implementation proposed in the process evaluation of complex interventions guide of the MRC.[17] We present broad features of the diverse contexts of each Parwarish implementation site. Evaluation of implementation presents quantitative and qualitative data summarising fidelity, dose and reach, followed by thematic analysis of the qualitative data presented under the domains of facilitation, community engagement and programme infrastructure. Mechanisms of impact are described within the qualitative thematic analysis.

### Context

Parwarish was implemented by teams from the Community Development and Health programme of the EHA in north India. The diverse study sites are summarised with a demographic profile of their three districts compared with national Indian data in table 1. While all three sites report high proportions of young people and high rates of gender-based violence, the Jharkhand site is predominantly tribal and rural, while Agra is urban with a marked asymmetry in the sex ratio. Only half of women in the UP rural site of Sonbhadra are literate and the majority live rurally. Gender inequality is a key contextual factor that influenced implementation across all sites, and this is evidenced in table 1. Asymmetries in the sex ratios of live births (fewer females) as well as the lower literacy rates of women underline the dominant patriarchy which is further discussed in qualitative findings of implementation.

The context is further described by considering the sociodemographic profile of community participants profiled in table 2. This demonstrates participants were highly disadvantaged and over 80% of participants were from scheduled tribes or scheduled castes, which are classifications assigned at birth and used by the Indian Government to indicate people groups who are structurally disadvantaged. Further indications of the social and economic disadvantage of participants are the fact that 55% of fathers and 75% of mothers had not completed primary schooling.

### Fidelity, dose and reach
#### Fidelity

Fidelity scores filled by facilitators and through each aspect of structured observation ranged from 70.0% to 82.0% (in a self-report five-item measure) and 75.7% to 78.6% (in a seven-item observed measure filled by coaches and researchers) suggesting fidelity to the intervention was moderate to good for all sites. Observers noted that facilitators initially omitted module components when pressed

**Table 1** Sociodemographic profile of the three districts where Parwarish was implemented, with reference data for India nationally

| Indicator | National— India | Agra city, Uttar Pradesh | Sonbhadra district, Uttar Pradesh | Khunti district, Jharkhand |
|---|---|---|---|---|
| Total population (million people) | 1200 | 4.41 | 1.86 | 0.53 |
| Population rural (%) | 72.2 | 54.2 | 83.1 | 91.5 |
| Population under 18 years (state, %) | 34.9 | 33.7 | 33.7 | 33.5 |
| Population scheduled castes/scheduled tribes (%) | 25.2 | 22.6 | 14.9 | 77 |
| Sex ratio (female to 1000 males) | 940 | 868 | 918 | 997 |
| Literacy (% literate female) | 65.5 | 61.2 | 52.1 | 53.7 |
| Literacy (% literate male) | 82.1 | 80.6 | 74.9 | 74.1 |
| Prevalence of gender-based violence (% at state level) | 40 | 47 | 47 | 30–50 |

**Table 2** Sociodemographic profile of Parwarish participants

| (N=239 families) | Agra (%) | Robertsganj (%) | Khunti (%) | Parent | |
|---|---|---|---|---|---|
| | Adolescents/household | | | Mother | Father |
| Female | 68 (52.7) | 36 (60.0) | 32 (64.0) | – | |
| Age in years (mean) | 13.3 | 13.3 | 14.0 | 36.8 | 40.2 |
| Caste (Indian census classification): | | | | | |
| Scheduled caste or tribes | 110 (85.3) | 51 (85) | 50 (100) | 199 (83%) | |
| Other backward castes | 19 (14.7) | 9 (15) | – | 37 (15%) | |
| General | – | – | – | 3 (1.3%) | |
| Religion | | | | | |
| Muslim | 11 (8.5) | 3 (5) | – | 14 (6%) | |
| Hindu | 117 (90.7) | 57 (95) | 3 (6) | 167 (70%) | |
| Christian | – | – | 21 (42) | 29 (12%) | |
| Other religion | – | – | 26 (52) | 29 (12%) | |
| Education | | | | | |
| Never been to school | 31 (24.0) | – | – | 135 (56%) | 83 (35%) |
| Class 5 completed | 46 (35.7) | 32 (53.3) | 23 (46) | 48 (20%) | 48 (20%) |
| Class 10 completed | 44 (34.2) | 48 (46.6) | 20 (40) | 43 (18%) | 83 (35%) |
| Class 12 completed | 8 (6.2) | – | 3 (6) | 2 (1%) | 7 (3%) |

for time and read role-plays aloud rather than acting them out. Review of the value of role-plays after module 5 led to improved fidelity in delivery of this component of the intervention. The key omissions in fidelity to documentation was in failing to write detailed comments in registers to critically reflect on sessions (challenges, highlights) in the first 5–6 modules. After review and discussion with facilitators this improved. Session duration was a mean of 1.5 hours (Khunti), 2.2 hours (Agra) and 2.8 hours (Robertsganj) with sessions in Khunti reportedly shorter due to the pressure of agricultural work in the Jharkhand setting.

### Dose and reach
There was 100% attendance in the Parwarish training by coaches and facilitators. Participant evaluations of Parwarish training were very positive, noting that there was an opportunity to rehearse and also put into practice the components of the intervention. For example, training facilitators took turns leading the morning 'emotional check in' and each also worked in pairs to develop their own household budget, a key output of later modules. Attendance of facilitators and coaches at weekly coaching meetings was over 90%. Attendance and engagement of participants in Parwarish are summarised below in table 3.

Participation was higher for women and adolescents than for men. Fathers explained their higher attendance in Agra due to encouragement by male facilitators so that just 13 of 129 fathers were classified as programme dropouts (missing four consecutive sessions), while few fathers attended in Agra and Khunti, reflecting dominant gender relations where men are expected to generate income.

### Implementation
The seven themes that emerged are presented under the domains of relevance, acceptability and feasibility linked to implementation of the Parwarish intervention and are summarised in table 4. A verbatim theme heading is provided to give greater texture to themes and is further illustrated with verbatim quotes taken from interviews and FGDs.

### Relevance
'They can watch if our words ring true'—facilitation by community members increased contextual validity of Parwarish. Overall, 3 coaches and 19 facilitators implemented Parwarish, and all (except 1) were parents of an adolescent and lived in the target communities. Every facilitator reported that arguments with shouting or hitting occurred in their own household illustrating shared lived experience with participants. The 22 facilitators and

**Table 3** Summary of participation (attendance) in Parwarish

| | Mean sessions attended (out of 14) | Mean number of home catch up sessions delivered (for those who missed a session) (out of 13) | Mean number of home activities completed (out of 13) |
|---|---|---|---|
| Mother | 11.6 | 1.1 | 9.8 |
| Father | 5.8 | 3.3 | 4.8 |
| Girls | 11.2 | 1.0 | 10.5 |
| Boys | 11.1 | 1.1 | 10.2 |

**Table 4** Summary of key themes

| Domain | Theme | Verbatim theme heading |
|---|---|---|
| Relevance | Facilitation by community members increased contextual validity of Parwarish | They can watch if our words ring true |
| | Implementation affected and was affected by gender relations | So, then we decided to take off our veils |
| Acceptability | Programme responsiveness to community needs increased participation | We talked about things I wanted to know |
| | Participation gathered momentum during implementation | I didn't plan to, but I ended up coming every week |
| Feasibility | Training promoted a strong core for implementation | We adjusted what we did depending on each group and each day |
| | Implementation was enhanced by technology | I liked seeing WhatsApp pictures to know what they were doing in Agra too |
| | Programme quality was strengthened by structured critical reflection | My husband and I sat and talked about what went well after facilitating each week |

coaches had a mean age of 39.9 years and 17 of 22 (77%) were from a scheduled caste or tribe reflecting a similar demographic makeup to participants.

Parents and adolescents described a high level of trust with facilitators at two levels; first, through pre-existing relationships with the implementing teams and second, through facilitators who responded to their current needs and issues. One family described how the Parwarish facilitators supported them when their adolescent self-harmed and needed hospital care. A facilitator explains how their community base increased connection and trust below:

Because we live in this place anyone can see what work we do and how we do things all day long. If someone comes from outside, then people don't know how that person really lives but if we live in the same place, they can see if our words ring true in how we live. So, I think it is good to live in the same place as group members and also, I know the best way to explain things as I know about them and their lives. SSI, Facilitator, Agra

Programme facilitation by community members also presented challenges: community facilitators described feeling scrutinised by their neighbours in their own parenting and communication ('they watch to see if our words ring true') and some struggled to observe professional work ethics, for example, to avoid gossip and breaches of confidentiality. Facilitators also described less engagement from groups most proximate to their own homes as described below:

Being a facilitator in my own village and among people of my caste is more challenging () as they don't take my words seriously and that reduces impact. But in other castes and communities people respond to my words positively. Facilitator, FGD, Robertsganj

This mistrust was somewhat mitigated by the team taking pains to be punctual, reliable and prepared and increasing trust leading to increased participation as Parwarish progressed.

'So, then we decided to take off our veils' - Implementation affected and was affected by gender relations

The context of patriarchal gender relations which value men and disadvantage women impacted implementation in numerous ways. Implementers described reduced participation from men and boys who were less available due to income generation responsibilities. The lower rates of literacy for women also led to difficulties in identifying women as facilitators. Gender norms dictated women are commonly responsible for nurturing aspects of parenting meaning many households 'assigned' attendance of sessions to mothers.

A positive aspect of the gender norms which limited women from doing paid work in Agra was that mothers and daughters were available for Parwarish sessions during the day although in tribal Jharkhand both men and women work full time meaning reflecting tribal communities practice perhaps more equal gender relations than mainstream north India.

The sessions were facilitated in a way that sought to overcome typical age-related and gender-related restrictions that might limit participation by young women, a key mechanism to support outcomes. Facilitators described how they sought participation from all with no 'right' or 'wrong' responses. A young woman described how this gave her courage to speak up:

Before this Parwarish course I would never give my opinion in front of adults I didn't know. But Anupriya (woman facilitator) would ask me to say what I thought and she said that there was no right or wrong answer so I could just give my opinion. Then I shared my opinion nearly every session and also I started to share my opinion at school. Young woman, Agra, FGD

Some resistance to traditional gender relations was evidenced through group participation which facilitators thought was triggered by both role modelling by facilitators as well as the process of equal valuing of participation by group members. For example, in Robertsganj and Agra, initially there was widespread practice of pardah (a gendered practice prevalent through north India dictating men and women related to each other should not sit together and women should hide their faces from older male family members). After several sessions, facilitators and several women decided to challenge status quo gender relations by sitting beside males and unveiling their faces as described:

> I decided it would be improper for men and women to sit together and see each other so I also wore pardah for first two sessions. But then I said I also couldn't understand what other women with faces covered were saying. So then we all decided to take our veils off and we were all showing our eyes. ()After two or three sessions in our facilitator group we also discussed the practice of purdah and use of face coverings and we decided we could make our own new rules for this group. SSI, Female facilitator, Agra site

### Acceptability

'We talked about things I wanted to know'—programme responsiveness to community needs increased participation. The interactive format of Parwarish was identified as core to engagement, with sessions described as participatory and non-didactic. Facilitators described role modelling the ideas as a key mechanism supporting outcomes, for example, ensuring women and men participated in discussions while discussing the importance of family decision-making. Parents described sessions were responsive to their concerns as father describes below:

> I really liked coming to the group because we'd talk about things we wanted to know about and everyone gave me suggestions to solve all these problems my wife and I were having. Then after that we would also discuss about the parenting thing and that was interesting so then I decided to stay for all the sessions. Father, FGD, Agra

Role-plays also provided participants with 'positive' and 'negative' examples of how to engage in family negotiations and communications about relevant issues to this community and were identified as a key mechanism to provoke thought and attitude change. A young woman described how the role-play approach helped her discuss her hopes for study with her parents:

> I was really wanting to talk with my parents who had said I should stop attending school, but I didn't know how. Then in one of the role plays about Nagma, she discusses with her mother what she really wants to do. I really loved that role play and then I felt I could

discuss with my mother about my hope to do more school study. Teenage girl, FGD, Agra

However, facilitators also described challenges in using a non-didactic approach. They argued that eliciting responses made sessions lengthy and at times they then felt there was no time to complete role-plays. However, the home activities (such as praising children or developing a family budget) helped keep families engaged. One mechanism of learning was that young people described that they would revise and teach the subject matter to their siblings in their households. A mother also described how she would reteach the weekly topic to remote family members by telephone as described below:

> I really love the plays and stories in the Parwarish sessions and so once I brought along my sister Mira who was visiting from Delhi. Later Mira wanted to learn more so I would ring her after each session to tell her everything that happened. Then Mira's mother-in-law in Delhi had noticed Mira was being appreciative and praising her often which was a new thing, so her mother-in-law also wanted to learn about the session, so now Mira's whole household listens to the phone updates when I ring them each week. Mother, FGD, Agra

'I didn't plan to but I ended up coming every week'—participation gathered momentum during implementation. Participants described multiple positive feedback loops that increased participation, engagement and ultimately outcomes as Parwarish progressed. Positive factors described included enjoyment of the group role-plays and games, a sense of growing friendship with other participants, hearing others stories of similar struggles and appreciating new ideas in content of sessions. Parents described planning to attend just one session but then attending weekly as described below.

> I really liked the idea of learning more about how to parent my children but I felt like I didn't have time for this, so I thought I'll just go twice and stop. But after two sessions there were so many funny stories and examples from other parents having exactly the same problems as we have. So, then I decided that I would come every time. Father, FGD, Agra

Later in Parwarish men described taking missing their daily wage work to participate and meeting at times of day they initially said were inconvenient. Momentum was also evident when adolescents, younger children and other ineligible community members requested if they could join the next programme.

### Feasibility

Implementation of Parwarish in these low-income settings was highly feasible because the programme required minimal physical infrastructure: private homes and courtyards were used for group meetings although this led to

challenges as homes were often too small for large groups impacting participation.

'We adjusted what we did depending on each group and each day'—training promoted a strong core for implementation

Facilitators described that the in-depth training and coaching provided a clear core intervention on which they made pragmatic microadaptations as a more porous periphery to the intervention. Facilitators and coaches described applying Parwarish learnings to their own household leading to positive changes which increased their engagement and provided examples they used in group sessions. Microadaptations described by facilitators were in response to community requests or contextual and pragmatic such as changing the reducing group size from 15 to 10 families to fit with crowded urban settings or holding sessions in late evenings in villages where agricultural work dominated daytimes.

'I liked looking at WhatsApp pictures to see what they were doing in Agra too'—implementation was enhanced by technology

Coaches and facilitators identified social media platform WhatsApp as central for communications, coordination, supervision and feedback in real time. Photos and text were shared many times daily to coordinate and facilitate problem-solving between implementers. For example, photos of groups in cramped small rooms led to the solution proposed to meet in a more spacious outdoor courtyard. Coaches and coordinating team members described how WhatsApp messages provided encouragement and they identified a sense of competition and ability to showcase work as a mechanism that promoted them to work harder as facilitators. For example, the Agra team described how they felt support from others via WhatsApp when they had to run groups in a flooded courtyard so they persisted despite inconvenience. Messages also were used for coordination and data collection, such as to update baseline research forms filled, which motivated other sites, and increased compliance and participation for all teams.

'My husband and I sat and talked about what went well after leading each week'—programme quality was strengthened by structured critical reflection. Parwarish implementation used formats and structures developed in PLH-Teen in South Africa to trigger critical reflection and improve intervention quality. For example, each facilitator filled a format after sessions describing highlights (what had gone well) as well as challenges (what did not go well) and their responses to observations and these were discussed with their coach. A fortnightly meeting between coaches in the three sites, the coordination team and the PLH-Teens support team used Skype group conversations with a translator increased the quality of coordination, reporting and real-time responsiveness and reduced travel.

Facilitators described how the formational Parwarish training changed their own attitudes to gender equality in marriage relations and adult–child (age) hierarchies. Coaches also described the fortnightly reports as improving their own group facilitation prompted them to help facilitators solve their own problems. An example of attitude change leading to behaviour change is described below:

(In our weekly facilitator meeting) Sushila described how she had never asked her three sons to help in the kitchen, but the other facilitators described how their sons would make chapatis and all. () So then Sushila told us how once getting late and so she had rung her son to ask him to wash dishes and he did that. She said it was a very positive experience.

## DISCUSSION

This process evaluation shows good acceptability and relevance for the Parwarish intervention evidenced by high levels of attendance, participation and engagement by parents and adolescents and implementation that was feasible in three disadvantaged urban and rural settings in central and north India.

Enrolment and attendance rates for fathers and mothers in Parwarish were significantly higher than in PLH-Teens implementations in South Africa, Philippines and Sudan.[12 25 26] Factors likely to have increased attendance include context-relevant adaptations by the community-based facilitators and the pre-existing strong relationships and community development work of implementing teams. The novelty of a first-ever research project with daily WhatsApp group reporting across sites perhaps added a competitive component that promoted greater activity among facilitators.[27]

The results demonstrate Parwarish was feasibly implemented by lay community facilitators and coaches. This has positive implications for scaling across South Asia and beyond: implementation by local people receiving modest remuneration in meeting spaces provided by the community makes this a relatively low-cost intervention.[25] However, while the programme is likely to be feasible and acceptable in other disadvantaged settings in South Asia, the high level of commitment by facilitators (eg, being willing to conduct meetings at night in Jharkhand communities) may be harder to replicate. Implementers also underlined the value of structured, high-intensity programme buttressing to increase fidelity and quality of Parwarish, which was also identified as central in PLH-Teens implementation[14 22 23] and other community-based group interventions in South Asia.[28–30]

Implementers and facilitators described the India-specific adaptation of PLH-Teens as relevant and acceptable in two key ways: first, the curricular adaptations that were superficial (eg, names of characters in role-plays) were appropriate and felt recognisable, while deeper changes, such as implementation by married couples and

holding group meetings in the private homes/courtyards in Uttar Pradesh sites, made Parwarish feasible, culturally appropriate and easy to engage with in these settings.[25 26]

Second, relevance and acceptability were increased by deployment of trained community members facilitating groups, using their in-depth knowledge of the daily lives, culture and language (ie, context) of participants. Parwarish facilitators used local dialects in facilitation and gave examples familiar to participants from shared common knowledge, adjusting logistics and role-plays in each setting to maximise participation. Implementers described the importance of trusting the innate contextual knowledge of facilitators[31] and the value of facilitators discussing their proposed adaptations to the programme with PLH-Teens coaches to ensure the core was followed but in locally acceptable ways. Facilitators described a range of continuous microadaptations, a feature key to successful interventions in other low-income and middle-income settings.[28 32 33] At the same time, some of the changes such as meeting at night perhaps reduced visual learning opportunities (eg, from role-plays) as there was no electricity at night.

The effectiveness of Parwarish was enhanced through the programme supports of coaching to increase facilitator competency, also a key theme in other PLH programmes.[23 25 26] Strategies increasing intervention fidelity and quality included observation and feedback by coaches, weekly meetings, checklists and fortnightly 'coach the coaches' sessions. The fidelity to the programme was acceptable but required structured feedback to solved problems such as meeting timing and needed attentive discussion to address more challenging aspects such as acting out role-plays which were omitted due to discomfort in initial modules. The need for a balance between fidelity and 'fit' with the local context is widely described[23 34 35] and in these findings there were some compromises in fidelity to 'fit' the community needs.

While participation of fathers in Parwarish was greater than has occurred in other PLH-Teens settings, men's attendance was still significantly less than women's attendance. Men's participation in any family linked intervention is critical as they typically control financial decisions and dominate family decision-making in Indian households.[36 37] Strategies to increase participation of men include including men as facilitators, ensuring meetings are held at times when men can attend and developing momentum for men's participation by engaging with men who are 'positive deviants'.[36] India is among the world's bottom five countries in the health and survival category of the Global Gender Gap Index meaning there are large gender disparities that disadvantage women.[6] Parwarish held transforming gender relations as core to process, content and implementation with men and women seated together in a group which already confronted the predominant unisex socialising and gender relations, an approach endorsed elsewhere in north India.[36]

Gender relations influenced participation, for example, decreasing attendance of fathers and sons, because of gender role expectations to generate income.[12 25 26] There were also proportionately more girls among participating adolescents, perhaps reflecting their lower priority for school attendance and employment.[6] Parwarish enabled new attitudes and transformations to gender relations. For example, the Agra and Robertsganj groups' decision to not observe purdah when in group meetings was key to the quality and quantity of dialogue in the group meetings and went on to lead to other outcomes where women described increased autonomy in decision-making and increased participation in family financial decisions. This early outcome generated a sense of entrenched behaviours linked to harsh parenting also being open to negotiation and change and encouraged facilitators, mothers and young women, which then increased their engagement and participation; this positive feedback loop has been identified as important in other gender transformative interventions.[36] Addressing deeply embedded gender relations of patriarchy prevalent across South Asia has no quick fix solution, and further work on gender transformative interventions is urgently needed in communities, schools and national policies across India to bring greater gender equality which can benefit both women and men.[36]

Parwarish groups used face-to-face trainings and meetings, and facilitator manuals and monitoring formats that were paper based, yet implementation was strengthened through technological applications and mobile phones. In India there are over 1 billion users of mobile phones[38] and 400 million people using the application WhatsApp.[39] The benefits of WhatsApp, which are familiar (widely used by peers), free and synchronous and easy to use, to support problem-solving and quality of community health programme implementation in low-resources settings have been described by others[27 40] and further realised in the COVID-19 era.[41 42] Facilitators and coaches described three new ways by which technology enhanced their work: first, in WhatsApp group communications between coaches and facilitators allowing sharing of pictures and video calls which strengthened monitoring and programme quality. Second, for coordination with implementers in New Delhi, with audio and visual facets increasing the ability to communicate the challenges and opportunities in field settings, and third, in the use of Skype conversations for 'coach the coaches' sessions between field sites, New Delhi and South Africa.

## Methodological considerations

There are some important limitations to this study. First, researchers were involved in higher level implementation of this project (PS and PN) which could have led to social desirability bias in responses from both facilitators and Parwarish participants. However, a benefit of including implementers as researchers was that PS, KM and PN had an in-depth understanding of factors influencing implementation as well as contextual challenges. Second,

the evaluation did not include a control group and was evaluated immediately after implementation limiting knowledge about sustainability of changes. Third, while researchers visited implementation sites several times, no researchers resided in intervention communities which reduced their contextual understanding of the relevance and acceptability of the implementation process. There were important strengths in the study design: First, it reflected the 'real-world' allowing iterative microadaptations of the intervention to make it optimally relevant and acceptable in the community. Second, implementation by community members as facilitators, with groups housed in communities using local homes to host groups and a modest budget in three diverse sites, provides a model for implementation that can be generalised for implementation and scaling by others working among people from low-income communities across South Asia.

## CONCLUSIONS

This evaluation demonstrates the Parwarish intervention as acceptable, relevant and feasible in an urban slum and a rural agricultural town in north India and a remote tribal region of central India. In a vast country which has seen almost no examples of programmes seeking to increase positive parenting and reduce family violence, this pilot study shows the implementation process was engaging, feasible and acceptable in communities, which paves the way for larger-scale implementation in other settings in India and South Asia.

**Acknowledgements** The authors thank Jacob Gwal and Englina Ela Rani Tirkey (coach) with the Khunti team, Somesh Singh and Deepika Tikvah (coach) with the Agra team and Jyotsana Saha (coach) with the Robertsganj team for all their enthusiasm and diligent work implementing Parwarish and supporting with data collection for this study. The authors are also thankful for the support in adaptation of Parwarish to Jeet Bahadur, Prerana Singh, Amanda Raymond and the experts by experience group of mothers and others in the Community Health and Development Programme of the Emmanuel Hospital Association.

**Contributors** KM conceived of the study, designed it with PS and IG and wrote the first draft and is overall guarantor of this paper, PN collected data with PS and completed analysis with KM, PP and IG and all authors contributed to multiple drafts of the paper.

**Funding** This work was supported by the Swedish Research Council: 2017-05421.

**Competing interests** None declared.

**Patient consent for publication** Not applicable.

**Ethics approval** This study involves human participants and was approved by Emmanuel Hospital Association's Institutional Ethics Committee, New Delhi, as Protocol 191 in March 2019. Participants gave informed consent to participate in the study before taking part.

**Provenance and peer review** Not commissioned; externally peer reviewed.

**Data availability statement** Data are available upon reasonable request to KM (kaaren@eha-health.org).

**ORCID iD**
Kaaren Mathias http://orcid.org/0000-0002-9607-9459

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
