## [Reviewer comments · BMJ Open]

ARTICLE DETAILS

TITLE (PROVISIONAL)	Is the Parwarish parenting intervention feasible and relevant for young people and parents in diverse settings in India? A mixed methods process evaluation
AUTHORS	Mathias, Kaaren; Nayak, Prabhudutt; Singh, Pratibha; Pillai, Pooja; Goicolea, I

VERSION 1 – REVIEW

REVIEWER	Alves, Elisabete Institute of Public Health of the University of Porto
REVIEW RETURNED	04-Aug-2021

GENERAL COMMENTS	The study aims to assess the feasibility, acceptability and relevance of the Parwarish, a positive parenting intervention in three diverse settings in India. The study is interesting but I have some commentaries: 1. Strengths and limitations of this study:- The first sentence should be better explored as a strength or limitation.2. Introduction:- Introduction section could be more organized and focused on available data. I miss some data regarding parenting and positive parenting in India, the possible association between positive parenting and family violence.- In order to understand the use of the 3 settings and a strength, I need to understand why I should expect differences among the 3 settings, especially since in table 1 there not seem to be relevant differences regarding the prevalence of gender-based violence.- What the authors intended to add to the literature should be clearly stated. Is this just a study with a regional interest? I believe that the focus should be put on more disadvantaged societies and not just India.- I would prefer to see a clear objective instead of the research questions.3. Methods:- “Intervention adaptation” section of Methods is repeating previously presented data.- It is not clear for me how were the 245 families invited selected, recruited or invited. Also, how was the distribution of the number of participants across the 3 settings controlled?- Information regarding informed consent and ethical approval should be included in the methods section. I believe that the section “Stance and ethics” needs to be rewritten. I do not think that the description of the authors is important. I believe that the
---

	authors should only describe the main characteristics of the research team that may be relevant for the interpretation and discussion of results. 5. Results:  - Results are clear and well written. - I would like to see some expressions that reflected the experiences and opinions of the adolescents. 6. Discussion:  - It is difficult to talk about feasibility without and evaluation of the program sometime after its implementation. - The low participation of fathers is also a question that cannot be ignored and needs to be more explored, especially in a setting where man are the ones that can make the more relevant decisions regarding the family. - The authors should clearly state how can these data from India be applied to the rest of the world? Or do they have local interest, only? - The limitations section of the manuscript needs to be improved and the main influence of such limitation on the interpretation of results needs to be discussed. - Overall, the text would benefit from a revision of English language, in order to be easier to read.
--	--

REVIEWER	Rasheed, Muneera A University of Bergen
REVIEW RETURNED	06-Oct-2021

GENERAL COMMENTS	The manuscript is generally well-written. The study describes the process and feasibility evaluation of a parenting intervention for adolescents across three different sites. My feedback to strengthen the manuscript is below: Introduction I would rephrase the first objectives to 'how feasible was it to implement the Parwarish intervention?' I think the second objective can be merged into the first one as feasibility also covers acceptability. Methods Conceptual framework: I think this can be placed with analysis section. Also, definition of the specific domains of feasibility: practicality, implementation, adaptations, acceptability and demand are required. Setting: Table 1: What does scheduled castes/tribes mean? There is a wide variation for this variable. What does this mean for the study? Intervention adaptation and content: Is it possible to add a supplementary file detailing the intervention content? It makes it easier for the reader to understand what is being evaluated. Information about dosage, frequency and duration of session will be important to highlight. A separate section around implementation team would also help contextualize the findings. Data collection: The evaluation of fidelity is unclear. What was the checklist about and what was being assessed? Table 3 is important, but it is not organized enough, and one needs to make an effort to make a sense of it. What re the different stakeholders: Families, coaches and facilitators? What were the modalities: online and face-to-face? Can the table be revised to better reflect the process?
---

	I was unable to locate statement on ethics approval. Please consent procedure under data collection. Results The themes presented need to be crisper and not phrases. It may be more appropriate to present the results under the five domains of feasibility else it difficult to evaluate. Discussion This is an interesting comment. Can the authors elaborate “However, while the program is likely to be feasible in other disadvantaged settings in South Asia, the high level of commitment by facilitators (for example, being willing to conduct meetings at night in Jharkhand communities) may be harder to replicate.” The authors have not discussed why did they not choose to hire independent team to conduct the interviews. There is an element of social desirability and more so in rural communities, and probably greater when the interviewers are senior members. Once the results are revised, I can better comment on the discussion. In the current form of findings, its difficult to draw conclusions about feasibility.
--	--

VERSION 1 – AUTHOR RESPONSE

Response to reviewers by Authors

REVIEWER ONE: Mrs. Elisabete Alves, Institute of Public Health of the University of Porto

Comments to the Author:

The study aims to assess the feasibility, acceptability and relevance of the Parwarish, a positive parenting intervention in three diverse settings in India. The study is interesting but I have some commentaries:

1. Strengths and limitations of this study:

- The first sentence should be better explored as a strength or limitation.

RESPONSE – Thank you, this has been clarified as a strength

2. Introduction:

- Introduction section could be more organized and focused on available data. I miss some data regarding parenting and positive parenting in India, the possible association between positive parenting and family violence.

RESPONSE – Thank you. We have added this paragraph:

Positive parenting interventions can potentially reduce violence in two key ways: firstly by promoting parenting strategies which reduce physically or emotionally violent actions, and secondly by reducing violent or anti-social behaviour among children(1).

Parenting in India has not been studied widely and a large systematic review examining interventions to reduce harsh parenting or promote positive parenting in low and middle income countries could not identify any studies located in India (1). Studies examining parenting of adolescents in middle-class urban settings identify parenting is often authoritarian, extends until early adulthood, with a focus on educational attainment, and with a high level of monitoring of the social behaviour of their teenagers

(particularly young women)(2-4). However, we could find no interventional studies promoting positive parenting among disadvantaged adolescents in India.

- In order to understand the use of the 3 settings and a strength, I need to understand why I should expect differences among the 3 settings, especially since in table 1 there not seem to be relevant differences regarding the prevalence of gender-based violence.

RESPONSE – We have added two narrative sentences to underline the differences in the sites and explain the value of implementation in diverse settings:

While all three sites report high proportions of young people and high rates of gender-based violence, the Jharkhand site is predominantly tribal and rural, while Agra is urban with a marked asymmetry in the sex ratio. Only half of women in the UP rural site of Sohbhadra are literate and the majority in that district live rurally.

- What the authors intended to add to the literature should be clearly stated. Is this just a study with a regional interest? I believe that the focus should be put on more disadvantaged societies and not just India.

- I would prefer to see a clear objective instead of the research questions.

RESPONSE: Thank you – we have re-stated the aim of the study more clearly as an objective and underlined the value of this study for other low and middle income settings in the conclusions.

This study aimed to examine whether the Parwarish parenting intervention was feasible, relevant and acceptable for parents, adolescents and community facilitators living in three diverse communities in Central and Northern India , in order to establish the utility of this intervention for scaled implementation across other South Asian settings. .

3. Methods:

- “Intervention adaptation” section of Methods is repeating previously presented data.

RESPONSE – The repeated section has been deleted.

- It is not clear for me how were the 245 families invited selected, recruited or invited. Also, how was the distribution of the number of participants across the 3 settings controlled?

RESPONSE- We have re-phrased the detail on participant recruitment to address these concerns.

- “EHA community health and development team members with support from community volunteers recruited families in communities where they had been running community health and development programmes in the previous five or more years, by informing them about Parwarish and inviting participation. They held small community meetings informing about the proposed programme and invited any families who met the inclusion criteria and keen to participate to give their names and details to a team member. Inclusion criteria were:

- at least one adolescent (aged 12 – 18 years) in the household
- permanent resident in the target community
- at least one parent/ carer and one adolescent available for weekly sessions.

A total of 239 of 245 families invited agreed to join the programme (97.6%). Based on team capacity it was decided to recruit 60 families at the two rural sites and 120 families at the urban Agra site. “

Information regarding informed consent and ethical approval should be included in the methods section. I believe that the section “Stance and ethics” needs to be rewritten. I do not think that the description of the authors is important. I believe that the authors should only describe the main

characteristics of the research team that may be relevant for the interpretation and discussion of results.

RESPONSE – The Stance and ethics section are presented at the end of the Methods section which is where it is recommended to be located. Providing information about author demographic identity including reflexivity about how this may influence research data collection and analysis is proposed as best practice in qualitative research, as referenced in the COREQ guidelines so we believe we should retain this section as it is(5).

5. Results:

- Results are clear and well written.
- I would like to see some expressions that reflected the experiences and opinions of the adolescents.

RESPONSE: This is a helpful observation thank you. Several verbatim quotes of adolescents have been added to the results section.

6. Discussion:

- It is difficult to talk about feasibility without and evaluation of the program sometime after its implementation.

RESPONSE – thank you. We have noted this as a limitation of this study (NB now at time of write up post COVID we continue to hear positive stories of outcomes however the official data collection did not extend to this time)

- The low participation of fathers is also a question that cannot be ignored and needs to be more explored, especially in a setting where man are the ones that can make the more relevant decisions regarding the family.

RESPONSE – Thank you – we have addressed the importance of increasing participation by men in the Discussion section:

While participation of fathers in Parwarish was greater than has occurred in other PLH_Teens settings, men's attendance was still significantly less than participation by women. Men's participation in any family linked intervention is critical as they typically control financial decisions and dominate family decision-making in many North Indian households(6, 7). Strategies to do this include including men as facilitators, ensuring meetings are held at times when men can attend, and developing momentum for men's participation by engaging with men who are 'positive deviants'(6).

- The authors should clearly state how can these data from India be applied to the rest of the world? Or do they have local interest, only?

RESPONSE _ Thankyou – we have edited the section addressing scaling to underline that these findings are relevant widely, with a focus on South Asia.

- The limitations section of the manuscript needs to be improved and the main influence of such limitation on the interpretation of results needs to be discussed.

RESPONSE: Thankyou – we have expanded and strengthened the limitations section and referred to how these limitations could limit interpretation of results. The updated Methodological considerations section is pasted below:

Methodological considerations

There are some important limitations to this study. First, researchers were involved in higher level implementation of this project (PS and PN) which could have led to social desirability bias in responses. A benefit of including implementers as researchers was that PS, KM and PN had an in-depth understanding of factors influencing implementation as well as contextual challenges. Second, the evaluation did not include a control group and was evaluated immediately after implementation limiting knowledge about sustainability of changes, and if possible a future study would examine outcomes one year after completion of the intervention to assure whether changes are enduring.

Third, while researchers visited implementation sites several times, no researchers resided in intervention communities which will have reduced their contextual understanding which may have reduced understanding of the relevance and acceptability of the implementation process. There were important strengths in the study design: First, it reflected the 'real-world' allowing iterative micro-adaptations of the intervention to make it optimally relevant and acceptable in the community. Secondly, implementation by community members as facilitators, with groups housed in communities using local homes to host groups and a modest budget in three diverse sites provides a model for implementation that can be generalised for implementation by others working among people from low-income communities increases in low income communities across South Asia

- Overall, the text would benefit from a revision of English language, in order to be easier to read.

Response: Thank you – we have proofread the entire manuscript and made edits to improve fluency and flow.

Reviewer: 2

Dr. Muneera A Rasheed, University of Bergen

Comments to the Author:

The manuscript is generally well-written. The study describes the process and feasibility evaluation of a parenting intervention for adolescents across three different sites. My feedback to strengthen the manuscript is below:

Introduction

I would rephrase the first objectives to 'how feasible was it to implement the Parwarish intervention?'. I think the second objective can be merged into the first one as feasibility also covers acceptability.

RESPONSE: Thankyou – I have re-phrased the objectives as aims, as suggested by Reviewer one and collapsed into a single objective as follows:

This study aimed to examine whether the Parwarish parenting intervention was feasible, relevant and acceptable for parents, adolescents and community facilitators living in three diverse communities in Central and Northern India.

Methods

Conceptual framework: I think this can be placed with analysis section. Also, definition of the specific domains of feasibility: practicality, implementation, adaptations, acceptability and demand are required.

RESPONSE: Thanks – this has been addressed in Analysis section as follows:

We also assessed feasibility which addresses facets of implementation that include practicality, pragmatic and implementation processes used to deliver Parwarish programme to participants, adaptations to the Parwarish intervention required to increase engagement, understanding and relevance to diverse communities. Additionally, we examined acceptability and demand for Parwarish by asking participants whether they were comfortable with the group implementation processes and programme content (8).

Setting: Table 1: What does scheduled castes/tribes mean? There is a wide variation for this variable. What does this mean for the study?

RESPONSE – Thanks, we have added a definition of this and provided further explanation as follows. “This demonstrates participants were highly disadvantaged and over 80% of participants were from Scheduled Tribes or Scheduled castes, which are classifications assigned at birth and utilised by the Indian Government to indicate people groups who are structurally disadvantaged.”

Intervention adaptation and content: Is it possible to add a supplementary file detailing the intervention content? It makes it easier for the reader to understand what is being evaluated. A supplementary file providing overview of the intervention has been developed and is pasted below:

Parwarish is a 14-module intervention adapted from PLH- Teens that is implemented in groups meeting for approximately 90 minutes weekly that are facilitated by a community facilitator that include approximately 10 parents and 10 adolescents. Each session starts with a period of exercises and emotional check-in (describing emotions and where they may be located in our bodies). The key content and title of each chapter is provided in the table below:

Session 1: Introducing the programme and setting participant goals
Session 2: Building a positive relationship by spending time together
Session 3: Praising each other
Session 4: Talking about emotions
Session 5: What do we do when we are angry?
Session 6: Problem solving: putting out the fire
Session 7: Motivation to save and make a budget with our money
Session 8/9: Dealing with problems without conflict I & II
Session 10: Establishing rules and routines
Session 11: Ways to save money and make a family saving plan
Session 12: Keeping safe in the community
Session 13: Responding to crisis
Session 14: Widening the circle of support

Information about dosage, frequency and duration of session will be important to highlight.

RESPONSE: We have moved the attendance and engagement data to give it higher priority in findings under the title of Fidelity, dose and reach.

A separate section around implementation team would also help contextualize the findings.

RESPONSE: We have added the following description of the implementation team: Parwarish was implemented by the EHA Community health and development programme teams based in Agra, Robertsganj (UP) and Khunti (Jharkhand). The budget for both implementation and evaluation was very limited and therefore we conducted the programme using in-kind support by Emmanuel Hospital Association existing resources and team members. A coach was appointed and trained for each location and took responsibility for recruiting facilitators as well as training and coaching facilitators. Separately, a project officer at each location was responsible for research components of the project and supported baseline and endline data collection as well as monitoring and evaluation of Parwarish sessions with other EHA community coordinators in the team.

Data collection: The evaluation of fidelity is unclear. What was the checklist about and what was being assessed?

RESPONSE: Thank you for these observations. We have extensively revised the methods sections to describe the intervention, recruitment and fidelity processes in greater clarity and detail.

Table 3 is important, but it is not organized enough, and one needs to make an effort to make a sense

of it. What re the different stakeholders: Families, coaches and facilitators? What were the modalities: online and face-to-face? Can the table be revised to better reflect the process?

RESPONSE: Thank you, we have re-formatted Table 3 to more clearly reflect themes and we have further clarified stakeholders and data collection processes in the methods.

Methods

In-depth interviews and focus group discussions (FGDs) were collected face-to-face in the community by KM, PS and PN in Hindi with 86 purposively sampled facilitators, parents and young people and continued until data saturation was reached. (). Further data was collected during meetings with coaches which were conducted online, in the form of written notes taken during meetings by a researcher not involved in meeting facilitation or participation (KM). Supplementary Table 1 provides an overview of the qualitative data collection.

I was unable to locate statement on ethics approval. Please consent procedure under data collection.

RESPONSE: Ethics approval statement is now provided at end of methods section.

Results

The themes presented need to be crisper and not phrases. It may be more appropriate to present the results under the five domains of feasibility else it difficult to evaluate.

RESPONSE: Thank you – we have reviewed the themes and re-labelled some of them to be crisper. We believe we have been able to give a more textured understanding of the themes by using verbatim quotes in addition to the narrative theme headings. This is a well-recognised process for qualitative studies which adds trustworthiness and detail (9) and therefore we would like to retain these. However, recognising your comment, we have re-ordered Table 3 reduce priority of verbatim theme sub-titles.

VERSION 2 – REVIEW

REVIEWER	Rasheed, Muneera A University of Bergen
REVIEW RETURNED	03-Jan-2022

GENERAL COMMENTS	The authors have made substantial revisions making the manuscript clearer for the reader. My only comment would be to map the themes to the evaluation outcomes (feasibility, acceptability, relevance).
--

VERSION 2 – AUTHOR RESPONSE

Dear Reviewer

I have completed minor revisions as suggested by Reviewer 2 and so now re-submit this paper.

Reviewer 2: The authors have made substantial revisions making the manuscript clearer for the reader. My only comment would be to map the themes to the evaluation outcomes (feasibility, acceptability, relevance).

RESPONSE: Thank you for this comment. We have revised the structure of the themes of the process evaluation of implementation and they are now grouped under the domains of feasibility, acceptability and relevance.

Please review the updated minor revisions in the submitted files.

Sincerely

Kaaren Mathias